# Comparison of Experimental Approaches Used to Determine the Structure and Function of the Class D G Protein-Coupled Yeast α-Factor Receptor

**DOI:** 10.3390/biom12060761

**Published:** 2022-05-30

**Authors:** Mark E. Dumont, James B. Konopka

**Affiliations:** 1Department of Biochemistry and Biophysics, University of Rochester Medical Center, Rochester, NY 14642, USA; 2Department of Microbiology and Immunology, Stony Brook University, Stony Brook, NY 11794-5222, USA; james.konopka@stonybrook.edu

**Keywords:** protein-coupled receptor, yeast pheromone receptor, yeast mating pathway, substituted cysteine accessibility method, disulfide crosslinking, second-site suppressor, constitutively active receptor, dominant-negative receptor, receptor oligomerization

## Abstract

The *Saccharomyces cerevisiae* α-factor mating pheromone receptor (Ste2p) has been studied as a model for the large medically important family of G protein-coupled receptors. Diverse yeast genetic screens and high-throughput mutagenesis of *STE2* identified a large number of loss-of-function, constitutively-active, dominant-negative, and intragenic second-site suppressor mutants as well as mutations that specifically affect pheromone binding. Facile genetic manipulation of Ste2p also aided in targeted biochemical approaches, such as probing the aqueous accessibility of substituted cysteine residues in order to identify the boundaries of the seven transmembrane segments, and the use of cysteine disulfide crosslinking to identify sites of intramolecular contacts in the transmembrane helix bundle of Ste2p and sites of contacts between the monomers in a Ste2p dimer. Recent publication of a series of high-resolution cryo-EM structures of Ste2p in ligand-free, agonist-bound and antagonist-bound states now makes it possible to evaluate the results of these genetic and biochemical strategies, in comparison to three-dimensional structures showing activation-related conformational changes. The results indicate that the genetic and biochemical strategies were generally effective, and provide guidance as to how best to apply these experimental strategies to other proteins. These strategies continue to be useful in defining mechanisms of signal transduction in the context of the available structures and suggest aspects of receptor function beyond what can be discerned from the available structures.

## 1. Introduction

This issue of *Biomolecules* is dedicated to the outstanding research career of Professor Jeremy Thorner. A hallmark of the research conducted by Professor Thorner and co-workers is the use of powerful yeast genetics and molecular biology approaches to explore important aspects of cell biology, particularly cell signaling, that are shared between yeast and higher eukaryotes. An outstanding example is the Thorner laboratory’s early work on the yeast α-factor mating pheromone and its processing, which has served as an important model for mammalian hormone processing [1]. Similarly, studies by the Thorner laboratory on the α-factor receptor Ste2p have served as an important model for understanding diverse aspects of G protein-coupled receptor signaling and are the subject of this review.

Haploid *S. cerevisiae* yeast cells occur in two different mating types, *MAT*α and *MAT***a**, that undergo mating to form a diploid cell [2]. Mating is initiated when peptide pheromones secreted by each mating type (α-factor or **a**-factor pheromones, respectively) bind to receptors on the surfaces of cells of the opposite mating type (Ste2p or the **a**-factor receptor, Ste3p, respectively). Most studies on pheromone receptor signaling have focused on Ste2p, as a result of difficulties in working with **a**-factor, the ligand for Ste3p, due to its modification by a hydrophobic prenyl group. *STE2* was first implicated as the gene encoding the α-factor receptor because *ste2* mutations rendered *MAT***a** cells (but not *MAT*α cells) defective in responding to pheromone [3], and because *ste2* mutant cells were defective in binding α-factor [4,5]. Thorner and co-workers provided definitive biochemical identification of Ste2p as the α-factor receptor by showing that Ste2p could be crosslinked to radiolabeled α-factor, and that heterologous expression of *STE2* in *Xenopus* oocytes resulted in production of a membrane glycoprotein that bound α-factor [6,7].

DNA sequence analysis of the *STE2* gene revealed that it encoded a protein with seven hydrophobic stretches, originally suggesting that it was a transmembrane protein, perhaps related to rhodopsin, which also has seven predicted transmembrane segments [8,9]. As more receptor sequences were determined, it became apparent that Ste2p and Ste3p were actually the second and third sequenced examples of the large extended family of G protein-coupled receptors (GPCRs) that each contain seven predicted transmembrane segments. The identities of Ste2p and Ste3p as GPCRs were subsequently confirmed by showing that activation of either receptor results in exchange of a bound GDP for GTP on the α-subunit (Gpa1p) of the cognate G protein, resulting in dissociation of Gpa1p from the dimer of the Gβ- and Gγ-subunits (Ste4p and Ste18p, respectively) [10,11]. The free Gβγ dimer then activates a MAP kinase cascade leading to cell cycle arrest, changes in transcription, and induction of chemotropic polarized morphological changes preparatory to cell fusion to form a diploid zygote [12,13]. There is also evidence for roles for dissociated Gpa1p in alternative signaling pathways [14], such as induction of autophagy [15]), and for some G protein-independent effects of pheromone receptors [16]. Furthermore, the *GPA1*, *STE4*, and *STE18* genes encode proteins with substantial sequence similarities to the α, β, and γ subunits of mammalian heterotrimeric G proteins [10,11,17]. In contrast, although Ste2p and Ste3p activate the same heterotrimeric G protein, they exhibit little sequence similarity with each other, or with mammalian GPCRs. However, the fundamental interactions between receptors and G protein are similar enough that GPCRs and G proteins can be functionally interchanged between yeast and mammalian cells [18,19,20,21]. Such interchangeability has been applied, in order to develop receptors for novel synthetic ligands, and for identification of novel GPCR ligands [22].

Ste2p shows an overall similarity to mammalian GPCRs in that the core region containing the seven transmembrane segments carries out ligand binding and G protein activation. Another similarity revealed by studies from the Thorner and Hartwell laboratories is that the cytoplasmic C-terminal tail of Ste2p, like the C-terminal regions of many mammalian GPCRs, is dispensable for signaling but important in regulating receptor desensitization and internalization [23,24]. More recent studies from the Thorner group and others have focused on the use of Ste2p for understanding mechanisms of receptor endocytosis and desensitization [25,26,27,28,29,30,31].

The extensive toolbox of genetic and molecular biological approaches that can be applied to yeast has been used to study Ste2p, leading to the identification of key structural features and functionally important residues. However, the lack of a three-dimensional structure of Ste2p to anchor these studies has limited the ability to understand Ste2p function. In spite of various efforts at purifying and crystallizing Ste2p for the last ~40 years [32,33,34], in addition to efforts to solve the structure using NMR [35], no high resolution structure of Ste2p was available until recently. A key breakthrough was the publication of the structure of a dimeric form of Ste2p in complex with the trimeric G protein that was solved by cryo-electron microscopy [36]. This was followed by an impressive collection of additional structures of Ste2p; in the unliganded state, bound to antagonist, and bound to agonist, all in the absence of G protein [37]. Taken together, these structures provide an important framework for understanding mechanisms of signal transduction by Ste2p. They also provide a unique opportunity, which we pursue in this review, to evaluate the wealth of data obtained from earlier genetic and biochemical studies against the newly-available structural data. Since a comprehensive review of the structure and function of Ste2p was published just prior to the availability of these structures [38], we focus here on the interpretation of previous results in light of the newly-available structural information, focusing on two questions: (1) How reliably did the biochemical and genetic approaches perform in anticipating the three-dimensional structure of Ste2p and explaining the receptor’s functional properties? (2) Can the comparison of the available structure with the results from the preceding genetic and biochemical approaches provide insights into dynamic properties and functional rearrangements of the structure that may not be evident from the cryo-EM analysis?

## 2. Topology

Initial predictions that Ste2p contains seven transmembrane segments with an extracellular N-terminus and an intracellular C-terminus were reinforced by detection of glycosylated residues in the presumed extracellular N terminus [39], by observations of phosphorylation and ubiquitination on the predicted C-terminal tail [24,40], and by patterns of protease accessibility of Ste2p in isolated membranes [41]. In order to gain more detailed information about the transmembrane topology of Ste2p, the substituted-cysteine accessibility method (SCAM) was used. This is a method in which individual residues in a protein are replaced by cysteines, and the accessibility of those cysteines for modification by membrane impermeant methanethiosulfonate derivatives are determined [42]. In the case of Ste2p, the commonly used reagent was MTSEA-biotin [[2-[(biotinoyl)amino]ethyl]methanethiosulfonate], which allows monitoring of the extent of reaction using probing blots with streptavidin-based detection systems. As shown in Figure 1, cysteine substitutions at 195 positions in transmembrane segments and loop regions of Ste2p have been tested for accessibility in this way [43,44,45,46,47].

There are a several obstacles to directly comparing results from SCAM to the cryo-EM structures of Ste2p [36,37]. One obstacle is that the cryo-EM structures do not provide any clear indication of the boundaries of the membrane, as the receptors and G protein-containing complexes used for structure determination were maintained in solution using the detergents lauryl maltose neopentyl glycol or glyco-diosgenin. Presumed lipid densities in the structures were tentatively modeled as cholesteryl hemisuccinate (CHS), which was present in excess during receptor purifications, apparently binding tightly enough to remain associated with the receptors through final sample preparation steps that did not include CHS. The minimal dimensions of the helical bundles in the direction perpendicular to the membrane are on the order of 40 Å, which corresponds roughly to the extent of the hydrophobic region of a typical membrane. However, in the ligand-free structure, transmembrane segment V (abbreviated as TM5) projects beyond the rest of the helical bundle in the cytoplasmic direction, and transmembrane segment I (TM1) projects out of the bundle into the extracellular space. Since cysteine accessibility measurements were generally performed in the absence of ligand, we focus here on the ligand-free structure PDB: 7QB9 [37].

As shown in Figure 1 and Figure 2, cysteine accessibility measurements resulted in a model of Ste2p topology that was in good agreement with predictions based on amino acid hydrophobicity [43], and turned out to be generally consistent with the relevant cryo-EM structure. At 9 of the 14 ends of transmembrane segments, high cysteine accessibilities (>50% of controls) were observed for positions that are within the helical segments identified in the structure, most likely because these positions are near or beyond the edge of the hydrophobic region. This is consistent with the observation that such accessible positions are found at five positions at the intracellular end of TM5 and at five positions at the extracellular end of TM1 that project beyond the ends of the rest of the helices in the structure. However, it is surprising that strings of up to seven accessible sites are found at the extracellular ends of the helical regions of TM3, TM4, and TM7, which do not project beyond the rest of the helical bundle. A likely explanation of this observation is that, as shown in Figure 2, all of these sites are adjacent to the α-factor binding site in the structure. Considering the fact that, in most cases, labeling was performed in the absence of ligand, accessibility to MTSEA-biotin may occur through an aqueous pathway provided by the unoccupied ligand binding site. In fact, decreased accessibility of cysteines substituted at a few tested positions (D124C, V125C, and Y128C in TM3) was observed when labeling was performed in the presence of α-factor. However, some positions in extracellular loop 1 (EC1) exhibited opposite behavior, increasing their accessibilities to MTSEA-biotin modification in the presence of α-factor. This presumably results from a ligand-induced conformational change in this region [46].

As shown in Figure 1, up to five accessible positions are also found at the intracellular ends of TM2, 4, and 6, which do not project beyond the helical bundle. In the structure of agonist-bound receptor in complex with G protein, the intracellular ends of TM2 and TM6, as well as the intracellular extension of TM5, are directly involved in the interactions with the C-terminal helix and non-helical extreme C-terminus of the Gα subunit (see Figure 2). Thus, this G protein binding site provides a possible aqueous route for modification by MTSEA-biotin. However, the route for labeling of four helical residues at the intracellular end of TM4 that are farther from the site of interaction with the G protein, is not clear from examination of the structure.

Unexpectedly, two regions of the receptor that are not immediately adjacent to transmembrane segments could not be labeled by MTSEA-biotin. One of these encompasses positions 107–121 in EC1. The inaccessibility of this region was originally interpreted as being indicative of a stretch of 3_10_ helix [46,49]. In the ligand-free cryo-EM structure of Ste2p, residues 108–114 and residues 116–120, in fact, form two short α-helices [37]. The inaccessibility of much of this region to modification is likely to be due to its close intramolecular packing against the extracellular portions of TM1 and TM4, and to intermolecular packing against the N terminus of the opposite monomer. An additional region with an alternating pattern of accessibility was identified encompassing residues 21–29 in the N terminus of the receptor [47]. This was originally interpreted as being indicative of β-strand structure and, as shown in Figure 2 and Figure 3, stretches of β-structure encompassing residues 23–28 are seen in the N-terminal regions of the ligand-free Ste2p structure. The alternating pattern of reactivity of residues in this region may result, at least in part, from intermolecular packing of the β-strands from the two monomers shown in Figure 3.

## 3. Helix–Helix Contacts

The use of compensatory and co-evolving amino acid substitutions in proteins to map and predict three-dimensional structures is now widespread [50,51,52,53]. The basic idea of such analyses is that the functional or structural effects of a change in the accessible surface or charge or hydrophobicity as a result of an amino acid substitution at one position may be reversed by a complementary change in a second position with which it is in direct contact. The approach is confounded by the many indirect mechanisms by which pairs of residues may interact, such as effects on global stability of cooperative folding, overall shifts in the positions of secondary structure elements, or in the case of receptors, overall shifts of equilibrium between inactive and active states. A commonly used criterion for identifying specific interactions between contacting residues is the requirement that the mutual effects of the amino acid substitutions be allele-specific, so that each mutation in the interacting pair provides compensation for the effects of specific substitutions at the partner position, but not at other positions in the protein. In principle, intragenic second-site suppressors can reflect either intramolecular interactions or intermolecular interactions between monomers of the affected protein.

Several studies used yeast genetic approaches to identify allele-specific intragenic second-site suppressors indicative of intramolecular interactions between different parts of Ste2p (see Figure 4A,B). Sommers and Dumont isolated starting mutations in the TM3 of Ste2p that resulted in reduced pheromone responses, then identified three second-site suppressors of these mutations [54]. One of the second-site mutations, the substitution of M218T in TM5 suppressed the loss of pheromone signaling of the initial mutations E143K and T144P in TM3, but failed to suppress other mutations in TM3 or mutations in TM1 and TM2. As shown in Figure 4, the site of the suppressor M218 resides immediately adjacent to the positions of the two starting mutations E143 and T144 in the Ste2p structure. Two other intragenic suppressors of the loss-of-function mutations in TM3 that were recovered in this analysis, Y266C and R58G, map to regions that do not contact the sites of the starting mutations, and proved not to be allele-specific for Ste2p signaling function [54,55]. It was ultimately found that the Y266C substitution by itself causes significant loss of signaling function, and second-site suppressors of this defect have now been recovered at widely distributed positions in Ste2p [55], demonstrating a lack of allele-specificity, and suggesting alternative mechanisms for the observed genetic interactions. Despite the overall lack of allele specificity of the R58G mutation for restoring signaling function, various substitutions for R58 were subsequently found to provide strong and specific suppression of the temperature-sensitive α-factor binding resulting from the substitution S95Y in TM2. This led to a proposal of contact between residues at these positions [34] that is consistent with the close proximity of R58 and S95 in the Ste2p structure (Figure 4).

An additional genetic interaction between residues in different transmembrane helices was identified between residue N84 on TM2, and Q149 on TM3 [56]. Substitutions at each of these positions suppressed constitutive activity and hypersensitivity to pheromone, resulting from substitutions at the other position (see Figure 4).

Another approach for detecting intramolecular interactions in the structure of Ste2p is the identification of intramolecular disulfides involving cysteine residues introduced at different positions in the protein (Figure 5A,B). Based on similarities between the phenotypes of mutations introduced at positions N205 at the extracellular end of TM5, and Y266 at the extracellular end of TM6, Lee et al., [57] postulated that these two residues might interact. Replacement of both of these residues by cysteine resulted in receptors that could not respond to α-factor. However, receptors containing these two cysteine substitutions, in addition to two constitutively activating mutations of *STE2* (P258L and S259L), retained constitutive signaling activity, hence raising the possibility that positions 205 and 266 might only interact in the activated state of the receptor. This possibility was reinforced by the observation that a disulfide could be efficiently formed between cysteines introduced at positions 205 and 266, but only when the receptors contained the constitutively-activating mutations. The structure of the agonist-bound receptor-G protein complex is consistent with an activation-specific interaction between N205 and Y266, since they are in close proximity with their side chains projecting toward each other (Figure 5C). In contrast, these side chains point in different directions in the presumably inactive antagonist-bound structure (Figure 5B).

Dube et al. [58] introduced pairs of cysteine residues with one each in TM5 and TM6, focusing on positions near the intracellular ends of these segments. Pairing of the substitution V223C in TM5 with substitutions of cysteine for L247, L248, or S251 all resulted in constitutive activation of the pheromone response pathway (i.e., enhanced ligand-independent signaling), despite the fact that none of the individual substitutions had this effect. The existence of a physical interaction between TM5 and TM6 at these positions was supported by the demonstration that a disulfide could be formed between cysteines introduced at positions 223 and 247. Constitutive signaling was also seen when the substitution L226C in TM5 was paired with M250C or S251C in TM6, however, this was not confirmed by disulfide crosslinking. In the cryo-EM structures, V223 and L226 consistently reside on one face of the TM5 helix facing TM6. On TM6 in the agonist-bound receptor-G protein complex, residue S251 is closely opposed to position 223 on TM5, however, other residues showing interactions with V223 and L226 on TM5 are arrayed at different angles with respect to TM5 (as dictated by local helical periodicity), and at different positions in the direction normal to the membrane (Figure 5).

Taslimi et al. created a library of randomly introduced cysteine substitutions in TM5 and TM6 of Ste2p, and screened for combinations of substitutions that resulted in constitutive activation of pheromone signaling that would be indicative of locking the receptor in an activated state [59]. They identified two pairs of substitutions resulting in constitutive signaling. One of these pairs the substitution N216C on TM5 with A265C on TM6. The other pairs the same N216C with F262C on TM6. Disulfide-linked peptides could be detected following digestion of the doubly mutated receptor with cyanogen bromide. Cysteine introduced at position 216 could also form disulfide crosslinks with cysteines introduced as replacements for residues S254, I261, I263, and L264 on TM6, although these had no obvious effects on signaling. Residue N216 on TM5 faces TM6 in the various solved Ste2p structures. However, the interacting residues in TM6 are pointed in different directions, and appear to be displaced towards the extracellular surface compared to the position of N216 (Figure 5).

Taken together, the detection of activation-dependent crosslinking and of constitutive activation resulting from the introduction of paired cysteine residues strongly suggested that relative displacement of TM5 and TM6 is involved in receptor activation. This is consistent with comparisons of activated and inactive structures of Ste2p showing a major change in the structure of TM6 upon activation; whereas TM6 exhibits a continuous helical structure in activated structures of the receptor, residues 252–255 of TM6 adopt a substantially rearranged non-helical conformation in the ligand-free and antagonist-bound structures. This change in the structure may reflect flexibility of this part of TM6 that could explain the observation that a small number of sites on one face of TM5 appear to be interacting promiscuously with a much wider range of positions on TM6, including positions that seem to be oriented incorrectly, or located too far away to directly contact the cysteine in TM5. In cases where the introduction of paired cysteine residues induced receptor activation, it is clear that the mutations must leave the receptor in a native-like conformation. However, for receptors containing paired cysteine residues that do not cause receptor activation but simply retain α-factor-dependent signaling, it is possible that crosslinking forces TM5 and TM6 into a non-native state; in this case, retention of signaling function in cells could be due to a residual population of receptors without crosslinks, since disulfide formation was never observed to be 100% complete.

Additional intramolecular crosslinking in addition to genetic interactions were reported between a cysteine substituted for Y26 in the N-terminal tail and cysteines substituted for V109 and T114 in EC1 [60]. This apparent crosslinking occurred both in the presence and absence of α-factor. Unlike other identified intramolecular disulfides in Ste2p, formation of this crosslink appeared to require treatment of membranes with the oxidizing agent Cu-phenanthroline. In the cryo-EM structures, Y26 is, in fact, located in close proximity to V109 and T114, as shown in Figure 5D for the agonist-bound receptor-G protein complex.

## 4. Ligand–Receptor Interactions

Genetic and biochemical approaches were extensively applied in order to map the sites of interaction of Ste2p with its physiological agonist, α-factor (sequence WHWLSFSKGEPMY) (for review, see [38]). Many mutations in both the receptor and ligand have been found to diminish the numbers or affinities of α-factor binding sites on the yeast cell surface. While effects of mutations on binding could be indicative of changes in the receptor that directly affect ligand interaction, they could also be the result of changes in overall receptor conformation, stability, biosynthesis, or subcellular localization. Thus, we focus here on experimental results that are indicative of specific ligand-receptor interactions.

Analysis of the binding, potency, and efficacy of different α-factor analogs led to the hypothesis that α-factor binds to Ste2p in a bent configuration in which residues 7–10 of the ligand adopt a β-turn-like structure promoted by the sequence Pro^8^Gly^9^. (Note that protein residues are labeled in one-letter code, and residues in the ligands are labeled in 3-letter code with residue numbers as superscripts.) Conformations of bound ligands were inferred from the effects of replacing residues in this region with d-Ala and l-Ala [61], and from the finding that derivatives of α-factor with chemically constrained conformations in this region retained significant biological activity [62,63]. Furthermore, monitoring of the effects on receptor signaling of amino acid substitutions and truncations in ligands indicated that the N-terminal residues of the ligand contribute both to binding affinity and to ligand efficacy, whereas the C-terminal region contributes primarily to ligand binding [38,64]. The newly-available structures of Ste2p [36,37] include structures in which the receptor is bound to normal α-factor and to the antagonist [desTrp^1^Ala^3^Nle^12^]α-factor with the sequence HALQLKPGQP(Nle)Y, where “Nle” refers to norleucine, used as a stable replacement for methionine that does not alter signaling properties [65]. The observed configurations of bound ligands in the structures of Ste2p clearly support the previous models of ligand conformations. Both agonist and antagonist bind with a turn-like bend facing the extracellular space and their N- and C-termini inserted more deeply towards the transmembrane region. In addition, the N-termini regions of bound agonist and antagonist exhibit strikingly different interactions with Ste2p. In the structure of the bound antagonist, the N-terminal residue His^1^ occupies a position that places its imidazole ring centered more than 10 Å from the position of the ring in the corresponding His^2^ of bound α-factor. This places His^1^ of the bound antagonist in a position that partially overlaps with the position that is occupied by Trp^3^ of the bound agonist (Figure 6).

Sites of contact of different regions of α-factor with particular groups in the receptor were previously studied by systematic mutagenesis of the receptor. For example, substitution of glutamic acid for Ste2p residues S47 and T48 enhanced affinity for an analog of α-factor containing lysine or ornithine as replacements for the normal glutamine at position 10, suggesting that the Glu^10^ in α-factor interacts with residues 47 and 48 of the receptor [66]. As shown in Figure 6, the side chains of residues S47 and T48 are directed towards the C-terminal regions of bound ligands in the antagonist- and agonist-bound structures; however, there is 8–9 Å of separation between them and the side chains of Glu^10^ in the ligands. (Note that residues Nle^12^ and Tyr^13^ are not modeled in the antagonist-bound structure).

In additional studies, substitutions at positions N205 and Y266 of Ste2p decreased affinity for normal α-factor, but increased affinities for forms of α-factor containing substitutions or truncations in the first four residues of the peptide, indicating that these residues at the extracellular ends of TM5 and TM6 interact with the N terminus of full-length α-factor [67]. These conclusions were supported by a thermodynamic mutant cycle analysis of interacting changes in affinity resulting from substitutions in the N-terminal region of the receptor and residues N205 and Y266 [68]. Figure 6 shows that there is a close interaction of Asn205 with residue Leu^4^ of bound ligand that is not significantly different in comparing the antagonist- and agonist-bound Ste2p structures. In contrast, residue Y266 of Ste2p is in close contact with Trp^3^ and Leu^4^ of bound α-factor in the active-state structure, but undergoes a reorientation that substantially reduces interactions with ligand in the antagonist-bound structure. These observations are consistent with effects of mutations of residues N205 and Y266 on receptor signaling (see below).

Further biochemical probing of α-factor-Ste2p interactions was conducted based on crosslinking of derivatives of α-factor containing groups that are chemically reactive [69,70] or photo-activatable [71,72]. An α-factor derivative containing a 3,4-dihydroxyphenylalanine (DOPA) group attached to Trp^1^ at the N-terminus of α-factor crosslinked to residue K269 in the EC3 loop of Ste2p. As shown in Figure 6, residue K269 is immediately adjacent to Trp^1^ of bound α-factor in the cryo-EM structure. In additional crosslinking studies, an α-factor derivative containing DOPA attached at the C-terminal position of α-factor normally occupied by Tyr^13^ crosslinked to Cys59 in TM1 of Ste2p. While this result is roughly consistent with the positions of the crosslinked groups in the Ste2p structure, the backbone-to-backbone distance between the Tyr^13^ and Cys59 is 14 Å, and Cys59 is two helical turns deeper into the transmembrane regions than the observed position of the C-terminus of α-factor (Figure 6B). Detection of a crosslink between residues that is not immediately adjacent may reflect mobility of the C-terminus of α-factor or of the attached dihydroxyphenyl group, or flexibility of the Ste2p TM1 region, allowing the reactive group on α-factor to preferentially react with the nucleophilic cysteine side chain in Ste2p.

Seven of the eight extracellular acidic residues in Ste2p could be replaced by uncharged residues with little or no effect on receptor function. However, replacement of D275 by asparagine resulted in pH-dependent changes in ligand binding and receptor signaling, suggesting that this residue undergoes an electrostatic interaction with α-factor [73]. In the structure of the agonist-bound receptor-G protein complex, the side chain of residues D275 is in direct contact with the side chain of residues His^2^ of α-factor (see Figure 6). Thus, alteration of the state of protonation of histidine near neutral pH may provide a possible explanation for the observed pH-dependent effects.

Another approach for mapping α-factor-Ste2p interactions used an α-factor derivative [Lys^7^(NBD),Nle^12^] containing the environmentally-sensitive fluorophore 7-nitrobenz-2-oxa-1,3-diazol-4-yl (NBD) [74,75,76]. Binding of this derivative to Ste2p places the fluorophore in a hydrophobic environment that blue-shifts and strongly enhances the intensity of the emission. These environmental effects on fluorescence were used as the basis for a genetic screen for mutations in Ste2p at residues that are in close enough proximity to the NBD fluorophore so that mutations at those positions specifically alter the wavelength of the fluorescent emission peak without significantly diminishing the receptor’s affinity for [Lys^7^(NBD),Nle^12^] α-factor. Mutations fitting these criteria were recovered at the extracellular ends of transmembrane helices at residues L44, N46, S47, Q51, I53, and F55 in TM1; at F99, Y101, and L102, in TM2; Y128 in TM3; F204, N205, and S207 in TM5; Y266 in TM6; and at Q272, G273, T274, and D275 in TM7. It is difficult to know the exact conformation of the fluorophore of [Lys^7^(NBD),Nle^12^] α-factor when it is bound to Ste2p, as the fluorescent group is too large to fit in the pocket occupied by the Lys side chain in the structure [77]. Thus, in evaluating proximity to the fluorescent group, we measured distance between the α-carbon of different mutation sites, and the α-carbon of the α-factor in the cryo-EM structure. Based on these measurements, as shown in Figure 7, all of these mutation sites identified in the fluorescence-based screen map to within 18 Å of Lys^7^ of α-factor.

## 5. Mutations That Constitutively Activate Receptor Signaling

Several groups have identified partially overlapping sets of point mutations in *STE2* that result in constitutive activation of the pheromone response pathway in the absence of the α-factor agonist (see Table 1) [56,58,78,79,80,81]. The availability of structures of Ste2p in unactivated (ligand-free and antagonist-bound) and activated (agonist-bound) states [36,37] now allows the evaluation of possible mechanisms of activation of receptor signaling by the identified constitutive mutations.

The strongest of the constitutive mutations are at residues P258, Q253, and S259 on TM6 (see Table 1, Figure 8). The mutation Q253L at one of these positions is located in a region of TM6 that undergoes drastic rearrangement from a non-helical to a helical structure in comparing ligand-free or antagonist-bound states, to activated agonist-bound states. This transition results in a ~6 Å translation of the α-carbon of Q253 that exposes the side chain to the external surface of the transmembrane helical bundle. Substitution of a leucine residue at this position may drive activation by favoring exposure to surrounding membrane lipids (Figure 8). A similar mechanism may be responsible for the weaker activating effects of the substitutions S254F and S254L at the adjacent position, where the effects of these less drastic hydrophobic substitutions for serine would not be expected to be as strong as replacement for glutamine. The other two sites of the strongest constitutive mutations are located just to the extracellular side of the helical transition in TM6 that results in a 38° shift in helix angle projecting toward the intracellular face. Despite the relatively small (2–3 Å) displacement of these residues upon receptor activation, mutations at these sites, particularly substitutions that remove or introduce proline residues, apparently propagate their effects toward the intracellular end of TM6, causing large displacements at the intracellular face that allow productive insertion of the C-terminus of Gα.

Several generalizations can be made concerning the extensive set of weaker constitutive mutations that have been identified (see Figure 8): (1) They are clustered in the intracellular half of the receptor helical bundle, generally facing the cavity into which the C-terminus of the Gα subunit is inserted. This indicates that none of the mutations act directly by mimicking the effects of agonist binding, perhaps because they are incapable of recapitulating the multiple ligand-receptor interactions that result in receptor activation. One exception that may in fact partially mimic ligand binding is the substitution N46S. (2) The constitutive mutations are found on all seven transmembrane segments. This suggests that there is not a single site of intramolecular interactions responsible for toggling between active and inactive states. Instead, activation may consist of a cooperative transition between states that can be driven in one direction or the other by perturbations at multiple sites that each contribute to the free energy landscape of the transition. One mutation that is difficult to understand in this context of the structure is I169K [56], at a site on TM4 that projects to the outside of the helical bundle in both inactive and activated states (Figure 8). Introduction of the polar lysine side chain on the surface of the helical bundle may lead to rearrangement of TM4 that promotes enhanced access of the intracellular surface to the G protein. (3) Most of the mutations are at sites that undergo small translations in comparing inactive and active structures of Ste2p. This may indicate the importance of hinge regions (as opposed to sites that undergo large activation-associated translations) in regulating the equilibrium between active and inactive states of the receptor. One such hinge region involves contacts between TM6 and TM7, where there are several residues that provide constitutive activation when mutated. Another such region is the string of contacts between the intracellular halves of TM3 and TM7, where sites of constitutive mutations line the helical surface of TM3 that faces TM7 (Figure 8C).

An intriguing aspect of the identified set of constitutive mutations is that many of the same mutations that confer constitutive activation also confer the ability to further activate Ste2p in response to binding of ligands that behave as antagonists toward normal receptors [59,64,80]. Another way of stating this is that the “antagonist” is actually a very weak partial agonist, and that the constitutive mutations increase the ability of the partial agonist to promote the transition to the activated conformation. This increased responsiveness was a general property of the constitutive mutants, even though responsiveness to antagonists was not part of the genetic screens used to isolate these mutations. One possible mechanism for the enhanced responsiveness to antagonists could have been that the mutations remove or alter interactions between the receptor and the N-terminal regions of bound antagonists that are known to control ligand efficacy. However, the available Ste2p structures reveal that none of the constitutive mutations are at sites that interact directly with ligand. Instead, the identified mutations appear to work in concert with the bound antagonist to shift the equilibrium of active and inactive states of the receptor toward the active state. The expanded volume of the ligand binding site and the reduced number of ligand-receptor contacts observed in the antagonist-bound structure, compared with agonist-bound structures [37], indicate that antagonist interactions with normal Ste2p may not be of sufficient strength to significantly shift this equilibrium towards activation. However, antagonist binding in conjunction with mutations that also contribute to shifting this equilibrium is apparently sufficient to provide detectable signaling responses. The broad distribution of mutational sites capable of providing this antagonist-to-agonist shift provides further evidence for cooperativity of the inactive-to-active state transition that does not depend on any single latch controlling the conformational shift.

## 6. Receptor Oligomerization

Prior to the availability of structures of Ste2p, evidence that Ste2p, like many other GPCRs, can form multimers was obtained through a variety of approaches, including the following: (1) fluorescence resonance energy transfer (FRET) [82,83,84,85] and bioluminescence resonance energy transfer (BRET) [86] between fluorescent proteins or bioluminescent proteins attached to the C terminus of Ste2p; (2) FRET between differentially labeled fluorescent α-factor- derivatives bound to the receptor [77]; (3) co-internalization of fluorescently tagged endocytosis-deficient (C-terminally truncated) and wild-type receptor alleles [87,88]; (4) bimolecular fluorescence complementation between fragments of fluorescent proteins attached to the C-terminus of Ste2p [89,90]; (5) intermolecular disulfide crosslinking; (6) co-immunoprecipitation of differentially tagged alleles of Ste2p [88]; and (7) functional intermolecular interactions between different receptor alleles (see below).

All of the available cryo-EM structures of Ste2p show the receptor as a dimer. In the single available structure of the receptor-G protein complex, each of the monomers of Ste2p is associated with a separate bound G protein trimer, though the structure was asymmetric; one of the G proteins was so poorly resolved that only the C-terminus bound to the receptor could be modeled. All receptor structures show that TM1 of Ste2p makes up the majority of the monomer-monomer interface, with an additional major contribution from the N-terminal tail (Figure 9). This agrees with several experimental approaches that were used to map intermolecular contacts involved in Ste2p oligomerization. The first of these used fragments of Ste2p that, when present with complementary fragments of the receptor, had previously been shown to be capable of reconstituting pheromone-responsive signaling function [91]. Based on FRET measurements and the abilities of the fragments to co-internalize with intact receptors, it was concluded that a major determinant of oligomerization was TM1, with some possible additional contributions from the N-terminal extracellular tail and TM2 [92]. A GxxxG sequence in TM1 (GVRCG in Ste2p) was identified that is similar to motifs implicated in helix–helix interactions mediating oligomerization of glycophorin and other transmembrane proteins [93]. Mutations in this motif diminished FRET between Ste2p versions fused to different fluorescent proteins, and reduced the efficiency of co-internalization of mutant C-terminally truncated and normal receptors [83]. Despite concerns that the tested mutations in the GxxxG motif only partially affect oligomerization (detected by BRET), and that the tested mutations severely affect receptor function and subcellular trafficking [86], the cryo-EM structure shows that the GxxxG motif figures prominently in monomer–monomer interface (Figure 9).

Dimer contacts involving TM1 are not only important for physical interactions between receptors, but also appear to play an important functional role in signaling. This is supported by the loss of function of receptors containing mutations in this region, as described above [83,86]. Consistent with this, an analysis of 576 random amino acid mutations in 143 positions in the transmembrane regions of Ste2p found only 24 positions exhibiting the most restricted range of conservative amino acid substitutions. Three of these positions form a contiguous stripe down one face of TM1 that can now be seen to map to the region of the helix comprised of the GxxxG motif involved in contacts between monomers [94].

Another approach that has been used for mapping intermolecular contacts between oligomerized receptors is the detection of intermolecular disulfides between receptors containing introduced cysteine residues. An interesting difference between these studies and the intramolecular crosslinking discussed above is that in most cases, detection of efficient intermolecular crosslinking required the addition of the catalytic oxidation-promoting agent Cu-phenanthroline, whereas most of the characterized intramolecular crosslinks formed spontaneously without the need to add an oxidizer. Consistent with the cryo-EM structures, cysteines introduced at two positions in TM1, V45, and V68, were reported to form intermolecular crosslinks between Ste2p monomers [95,96]. In view of the extensive intermolecular interactions between TM1 helices seen in the structures, it is surprising that no additional crosslinks involving TM1 were identified in these two studies, despite testing cysteine substitutions at thirteen additional positions extending across much of the transmembrane region of TM1. In some cases, this may reflect total loss of receptor function resulting from interference with dimerization caused by the cysteine substitutions.

Additional intermolecular crosslinks were also detected between cysteines introduced at positions in TM7 of Ste2p, including residues T278, A285, and L289, in addition to residues 291–296 [95]. The intracellular ends of TM7 do, in fact, approach within a few Ångstroms of each other in structures of Ste2p bound to α-factor, although their closest approach is about 20 Å in antagonist-bound and ligand-free structures; this change in proximity results from the substantial α-factor-induced rearrangement of the intracellular end of TM7. Thus, it is surprising that Kim et al. reported that intermolecular crosslinking of this region of TM7 is generally reduced in the presence of either α-factor or antagonist [95]. Furthermore, even in the α-factor-bound state, reported sites of crosslinking near the middle of the transmembrane region, such as T278 and A285, are 20–30 Å apart in all the available structures (Figure 9).

Additional intermolecular disulfides consistently have been reported for cysteines introduced into the Ste2p N-terminal extracellular tail at the positions of G20, S22, I24, Y26, S28, and Y30 [47,60] (see Figure 9C). For some of these sites, such as the I24C and Y26C substitutions, such crosslinking is compatible with all the available structures, as strands encompassing these residues from the two monomers are closely apposed in antiparallel orientations. (They are β-strands in the unactivated states that become less ordered in agonist-bound states, but the positions of the side chains remain similar.) However, it is harder to understand crosslinks between cysteines replacing residues G20 and Y30 at the outer limits of this region, since the antiparallel arrangement places the corresponding α-carbons of these residues on the two Ste2p monomers that are 23 Å (for G20C) and 30 Å (for Y30C) apart. The number and apparent promiscuity of disulfides involving the N-terminal region suggests that crosslinks form more readily in this region than in TM1, either because of more favorable chemistry in a less hydrophobic environment, or because the N-terminal tail is maintained in less a rigid conformation than TM1.

A set of reported disulfides that is difficult to reconcile with the available structures is the string of positions including V183, V186, K187, M189, and I190 at the extracellular end of TM4 [96], since the backbone positions for these residues are at least 27 Å from the nearest backbone positions on the opposite monomer (see Figure 9). These crosslinks raise the possibility that the oligomeric state of the protein in the solved structures might differ from that in native membranes, where the receptor may form higher-order oligomers involving contact sites in TM4. Alterations in oligomeric states could be the result of using detergent-solubilized Ste2p and Ste2p-G protein complexes that had been reconstituted from proteins expressed in multiple heterologous expression systems [36,37]. The existence of higher-order oligomers has been suggested by several publications [90,96,97]. However, there also remains a possibility that some disulfides detected following oxidizing treatments may result from partial denaturation or aggregation of receptors.

## 7. Dominant-Negative Mutations

Some alleles of *STE2* exhibit dominant effects. For example, expression of wild-type Ste2p in the same cells as mutant receptors with constitutive and hypersensitive effects on pheromone signaling suppresses the effects of the mutant receptors [23,24,79,80,81]. In addition, certain mutant forms of Ste2p confer dominant-negative signaling phenotypes, meaning that the mutant receptor inhibits signaling by wild-type Ste2p expressed in the same cell. These mutations include the substitutions N132Y, N132I, Q135P, M180R, S184R, A185P, Y203H, F204S, N205K, N205D, S207F, L264P, Y266C, Y266D, T274A, and D275V [98,99]. As shown in Figure 10, in contrast to the constitutive mutations, the dominant mutation residues cluster near the extracellular ends of five helices (TM3-TM7). Although these positions are close to the α-factor binding site, the mutations do not simply act by blocking ligand binding, since the mutations with the strongest phenotypes have only modest, if any, effects on ligand-binding affinity [86,98]. Furthermore, not all loss of function *ste2* mutations are dominant. For example, the recessive mutation L236H in the third intracellular loop prevents G protein activation, but is otherwise capable of binding α-factor and being stimulated to undergo ligand-stimulated endocytosis [100]. The positions of the identified dominant-negative mutations indicate that they lock Ste2p in an inactive state by blocking ligand-induced conformational changes, such as the substantial shrinking of the volume of the ligand binding pocket noted in comparing activated to inactive structures [37]. Comparison of the active and inactive structures reveals that α-factor binding induces relatively small conformational changes in the extracellular half of the receptor that are amplified into much larger changes at the intracellular face. This apparently allows the single amino acid substitutions of dominant negative mutations in the vicinity of the ligand binding site to be effective in blocking receptor activation. In fact, it is generally observed that small changes at the extracellular ends of the α-helical bundles of GPCRs result in larger changes on the intracellular face, leading to G protein activation [101,102].

Two primary models have been proposed for the mechanism by which dominant-negative mutant receptors interfere with the function of co-expressed wild type Ste2p. One model is that the dominant forms of Ste2p associate stably with G proteins, thereby preventing other co-expressed alleles of the receptor from gaining access to G proteins needed to activate downstream signaling [100]. Some studies supporting this model were based on assays of dominant-negative mutants for their ability to maintain α-factor-induced cell division arrest for two days, which requires a high level of Ste2p on the cell surface.

An alternative model is that dominant effects result from cooperative conformational interactions among oligomerized receptors. If receptor activation requires a cooperative interaction between dimerized receptors, it is likely that mutations that block activation-associated conformational changes in one receptor could also block signaling by co-dimerized wild-type receptor. Using a short-term reporter gene assay to monitor pheromone signaling, evidence was obtained that oligomerization is important for the dominant effects of mutant forms of Ste2p [86,103]. Consistent with this, it is interesting that no dominant-negative mutations were found in TM1 or the C-terminal region of TM7, the regions closest to the Ste2p dimer interface, suggesting that maintenance of the dimer is important for the dominant-negative effects, and perhaps for maintenance of cooperative interactions.

The available structures of Ste2p reveal an unanticipated change in the dimer interface upon receptor activation, resulting in an increased number of ligand-receptor contacts, a significant increase in the surface area of interaction, a greater calculated energy of interaction, and a shift of the relative positions of the monomers that brings them 3 Å closer together. In addition, molecular dynamics simulations of the dimer revealed patterns of correlated residue movements that extend across the dimer interface. However, activation of pheromone signaling is not cooperative to the extent seen in the GABA_B_ receptor, where one receptor binds ligand and a separate co-oligomerized receptor activates G protein [104], since co-expression of a mutant form of Ste2p that cannot bind α-factor with another mutant form that cannot activate G protein did not result in pheromone sensitivity [105]; moreover, co-expression of certain inactive mutants of the receptor with normal Ste2p did not inhibit conformational changes in the normal receptor detected, based on protease sensitivity [87].

## 8. Conclusions and Future Directions

The conclusions from examination of the family of newly-available cryo-EM structures of Ste2p are generally consistent with the results of previous biochemical and genetic approaches for probing the structure and, in fact, this previous body of results formed a basis for some of the interpretation of the structures. To the extent that discrepancies were identified between previous approaches and the cryo-EM structure, these can be interpreted either as illustrating limitations of the previous approaches, or as illustrations of the possible usefulness of biochemical and genetic approaches in going beyond the static structure to illuminate dynamic and functional properties of the receptor.

Cysteine accessibility measurements provided a good overall view of the topology of Ste2p in the membrane, confirming the approximate locations of transmembrane segments and extra-membrane loops that were predicted based on sequence hydrophobicity. Since the cryo-EM structures were solved in detergent, the exact boundaries of transmembrane segments are not yet precisely defined. Furthermore, the α-helical structures of several transmembrane segments extend beyond the predicted boundaries of the transmembrane region in the available structures, complicating interpretation of accessibility to labeling from the aqueous milieu. The observed patterns of cysteine accessibility appear to be determined, in part, by aqueous pathways of accessibility involving the ligand binding site and site of interaction with the G protein α-subunit. The detection of inaccessible sites in the N-terminus and EC1 loop also led to correct conclusions that these regions were involved in secondary structures, and in intramolecular and intermolecular interactions; however, some of the conclusions about the particular types of involved secondary structure turned out to not to be in complete accord with the cryo-EM structure.

The characterization of second-site intragenic suppressors correctly identified several helix–helix interactions in Ste2p that were apparent in the cryo-EM structures. However, since there can be multiple mechanisms of suppression involving global effects on protein conformation and stability, it is clearly important to rigorously test for allele specificity if suppressor analyses are to be used to draw conclusions about structural contacts.

Intramolecular disulfide crosslinking of site-directed cysteine substitutions was a useful approach for structural analysis, and for identification of conformational changes associated with receptor activation, particularly considering the convenience of using starting forms of Ste2p that lack cysteine residues but maintain normal signaling responses. Detection of constitutive activation of receptors resulting from introduction of pairs of cysteine substitutions provided strong evidence for interactions between TM5 and TM6 that are associated with receptor activation. The biochemical identification of disulfides involving cysteine residues introduced into TM5 and TM6 also allowed for mapping of interactions that are not associated with receptor activation. However, the promiscuity with which particular sites on TM5 could crosslink to multiple positions on TM6 was surprising, perhaps indicating conformation flexibility or the dynamics of TM6. It is not clear whether the failure to identify disulfides involving cysteine residues in transmembrane segments other than TM5 and TM6 is a result of the experimental strategies used for targeting sites of cysteine substitutions, or if it might be indicative of special properties of the interaction between TM5 and TM6.

The cryo-EM structures confirmed considerable previous work that established a bent conformation for bound α-factor, and the existence of separate interactions of the N and C termini of α-factor with different regions of the receptor [38]. The structures also confirmed the proximity to the α-factor binding site of residues of Ste2p that, when mutated, change the optical properties of a bound fluorescent α-factor derivative [64]. It had previously been proposed that both the N and C termini of α-factor participate in binding interactions with the receptor, but the interactions of the N-terminal region are more directly involved in receptor activation [38]. This is consistent with the detection of very different interactions of N termini of the agonist and antagonist with TM5, TM6, and the EC3 loop of the receptor in the relevant Ste2p structures.

Together with disulfide crosslinking experiments, the locations of the strongest constitutively activating mutations in Ste2p implicate TM6 as a major determinant of receptor activation. However, the presence of additional weaker activating mutations in every transmembrane segment of Ste2p indicates that alterations in many different parts of the protein are capable of altering the distribution of states in a cooperative transition between inactive and active receptor conformations. The restricted localization of constitutively activating mutations to central and intracellular-facing portions of the receptor may reflect an inability of single mutations in the ligand binding site and other extracellular-facing regions to mimic the multiplicity of interactions by which α-factor binding activates the receptor. Since activation of the pheromone pathway leads to cell cycle arrest, Ste2p may have evolved multiple mechanisms to prevent pheromone responses in the absence of α-factor binding. Thus, instead of mimicking multiple interactions with agonists, constitutively-activating mutations may need to directly alter intracellular-facing portions of the receptor that interacts with the G protein.

Ste2p forms dimers under the conditions used for the cryo-EM structure determinations; however, questions remain about the receptor’s functional oligomeric state. Since cry-EM was performed on purified Ste2p in the presence of detergent and, in some cases, using a reconstituted complex with components purified from different heterologous expression systems, it remains possible that the resulting structures do not completely recapitulate the oligomeric state of Ste2p in yeast membranes. Results from genetic analyses and intermolecular disulfide crosslinking are in striking agreement with the observation of a dimer interface involving a GxxxG motif in TM1 in the cryo-EM structure. However, the detection of additional sites of crosslinking that involve residues nowhere near the structurally defined dimer interface raises questions with regard to the possibility of higher-order oligomeric states in native membranes, or the possibility of experimental limitations in crosslinking experiments.

The structures of Ste2p dimers in different activation states reveal surprising state-dependent differences in contact sites between monomers, in the extent of monomer–monomer interactions, and in relative positioning of monomers. However, the functional role of receptor homo-oligomerization in signal transduction by Ste2p, and for many other GPCRs, remains unclear. Genetic results, such as the negative effects of mutation of the GxxxG motif in TM1 [83,86], and the ability of dominant-negative forms of Ste2p to interfere with co-expressed wild-type Ste2p over a wide range of abundance of G protein subunits [103], are suggestive of a role for cooperativity between monomers in signaling. However, approaches to directly detect cooperativity between monomers in signal transduction have not succeeded. No cooperative effects have been observed in the experiments that monitored the dependence of ligand binding and activation of the receptor on α-factor concentration. Furthermore, it has not been possible to find pairs of defective Ste2p mutants in which binding of α-factor to one receptor in a dimer can trigger G protein activation by the other member of the dimer [105].

The determination of the family of structures of Ste2p and Ste2p-G protein complex constitutes a pivotal step for understanding the functioning of the pheromone-sensing machinery. Ste2p is of particular interest because of the maintenance of overall signaling function by activation of the heterotrimeric G protein, in spite of its extreme sequence divergence from mammalian GPCRs, and even from the other yeast pheromone receptor Ste3p, which acts on the same G protein trimer. However, the particular activation-dependent motions of TM6 and TM7 observed in the structures of Ste2p are notable, since they add to from the growing diversity of helical rearrangements that have been observed in other GPCRs [37,106].

The combination of cryo-EM with biochemical and genetic approaches can be expected to be useful in resolving critical outstanding questions about the mechanisms of signaling by Ste2p. These include the following:

(1) What is the mechanistic basis for receptor activation? Which of the conformational changes observed in the structure of agonist-bound Ste2p actually drive G protein activation, and which simply accompany the observed changes in state? One step towards addressing these questions would be to solve structures of multiple constitutively activated mutant forms of the receptor, in order to look for common elements. Another approach would be to use the available structures as a basis for predicting and creating an expanded and more complete set of pairs of cysteine substitutions capable of constitutively activating the receptor. An additional approach would use the available structures to design multiple amino acid substitutions that could stabilize the activated state of the receptor. Despite the fact that binding of agonists at the extracellular-facing surface of the receptor is sufficient to cause receptor activation, nearly all the previously recovered single amino acid constitutive mutations are located in intracellular portions of the structure. This suggests that multiple simultaneous changes in the structure and in the interactions of extracellular regions are required in order to activate signaling. Based on the newly available structural information, it should be possible to design multi-site combinations of activating mutations that would have been missed by previous genetic screens.

(2) Can new ligands and modes of ligand binding be identified and characterized? In addition to the extensive set of ligands that have previously been tested for interactions with Ste2p [107], the availability of detailed structures of the receptor in different states of ligand association raises the possibility of designing novel ligand and receptor variants that allow for new receptor–ligand interactions. For example, knowledge of the structure of the receptor inactive state could be used as the basis for designing new receptor antagonists. The commonly used α-factor derivative [desTrp^1^Ala^3^Nle^12^]α-factor is an N-terminally truncated form of α-factor that, by virtue of the truncation, fails to participate in interactions with the receptor that are required for activation of signaling. Thus, it is not surprising that, under certain circumstances, [desTrp^1^Ala^3^Nle^12^]α-factor behaves like a weak partial agonist [64,80]. We envision a new class of full-length Ste2p antagonists with substituted N-termini that, in contrast to [desTrp^1^Ala^3^Nle^12^]α-factor, would provide new contact sites that directly stabilize the inactive state of the receptor. It may also be possible to use structural information to design super-agonists with higher levels of ligand potency or efficacy compared to normal α-factor. It would also be of particular interest to conduct further structural studies in order to characterize the modes of binding and action of reported allosteric modulators of Ste2p signaling, such as novobiocin and peptide synergists [108]. Another interesting area for further investigation based on the new structural information is the process of co-evolution of mating pheromones and their receptors among divergent yeast species.

(3) What is the oligomeric state of Ste2p in native membranes, and how do monomer–monomer interactions affect signaling? Additional intermolecular cysteine disulfide crosslinking studies informed by the cryo-EM structures should be performed in membranes and use detergent-solubilized receptors to determine whether methods for receptor solubilization might affect the oligomeric state, particularly with regard to previously reported sites of crosslinking that do not map to the dimer contacts in the Ste2p structures. It will also be important to evaluate the role of cooperative interactions between dimerized receptors, such as those suggested by comparison among the different structures of Ste2p, and those reported for other receptor homo-oligomers [109]. This can be addressed by characterizing the functional effects of mutations at sites of differential monomer–monomer interactions uncovered while comparing active and inactive structures of Ste2p [37].

(4) Are there differential modes of interaction of Ste2p with G protein? Since incubation with purified G protein trimer was used to stabilize ligand-free Ste2p for purification and structure determination [36], it may be possible to solve a structure of the ligand-free receptor with G protein in order to address the important question of how receptor-G protein interactions vary between the ligand-free and agonist-bound states. It would also be informative to see whether different modes of receptor-G protein interaction can be identified by solving structures of constitutively-active and dominant-negative mutant forms of the receptor in complex with G protein. Since regulators of G protein signaling (RGS proteins) appear to interact directly with some receptors (including Ste2p) and receptor-bound G proteins [25,110,111,112], it would also be exciting to be able to determine the structure of a complex of Ste2p with G protein, and with the RGS protein Sst2p.

## Figures and Tables

**Figure 1 biomolecules-12-00761-f001:**
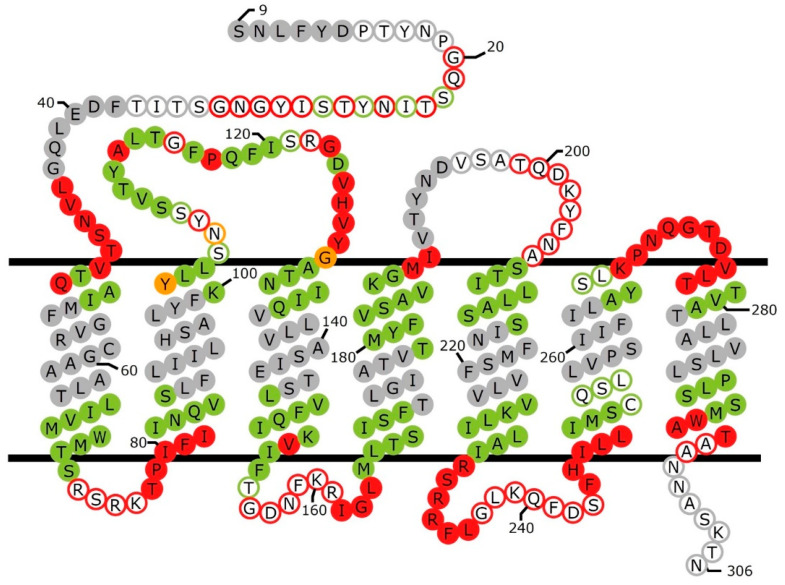
Schematic view of accessibility measurements and helical segments in Ste2p. Filled circles of different colors indicate positions of residues that map to α-helical regions of the ligand-free structure (PDB: 7QB9) [37]. Circles of different colors with white backgrounds show the positions of residues that do not reside in α-helical segments. Positions that, when replaced by cysteine, exhibited greater than 50% reactivity with MTSEA-biotin, compared to controls presented in the corresponding publications [43,44,45,46,48], are shown in red. Positions where cysteine substitutions were tested but exhibited less than 50% reactivity with MTSEA-biotin are shown in green. Positions that were not tested for cysteine accessibility are shown in grey. Positions labeled in orange were reported to have different accessibilities in different publications. Approximate boundaries of the membrane region are indicated. Positions of amino acids in the figure are not intended to correspond to their actual locations in the cryo-EM structure. The first eight residues of the N terminus and the last ~125 residues at the C terminus of Ste2p were not resolved in the structure, and are not included in the figure.

**Figure 2 biomolecules-12-00761-f002:**
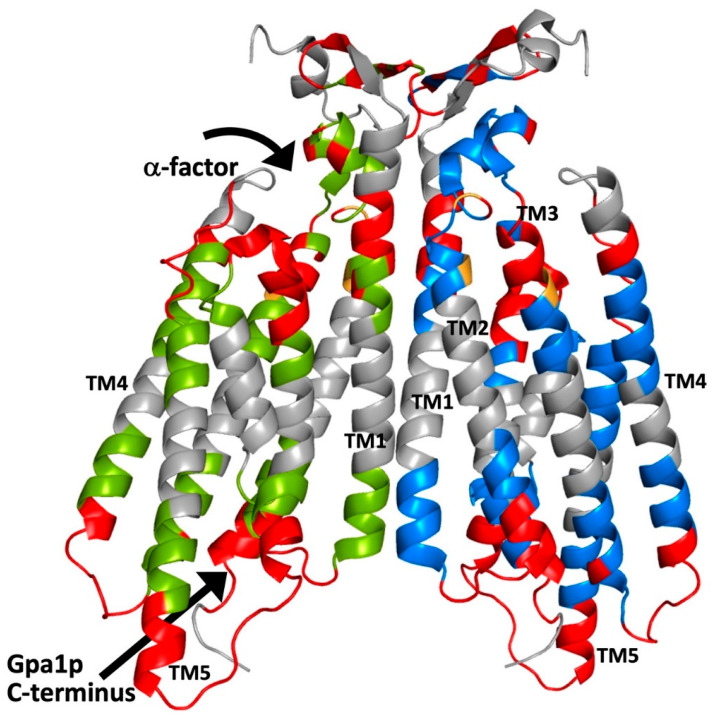
Amino acid positions in Ste2p that reside in transmembrane α-helical regions of the structure are accessible to labeling with MTSEA-biotin when mutated to cysteine. This figure shows the data from Figure 1 mapped onto the cartoon representation of the three-dimensional structure of the ligand-free Ste2p dimer (PDB code: 7QB9) [37]. Positions of residues that, when replaced by cysteine, were accessible to MTSEA-biotin are colored red. Positions that were tested but not accessible to MTSEA-biotin are colored green or blue for the two different Ste2p monomers. Positions that were not tested for cysteine accessibility are shown in grey. Positions for which there were discrepancies between publications are shown in orange. (All structural models presented in this manuscript were prepared using the PyMOL Molecular Graphics System (Schrödinger, LLC)).

**Figure 3 biomolecules-12-00761-f003:**
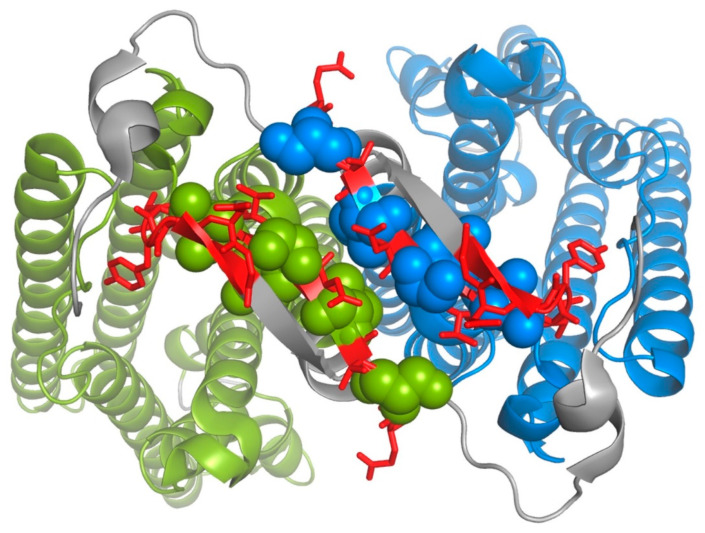
Accessibility of the N-terminal region of Ste2p mapped onto the structure of the ligand-free Ste2p dimer (PDB code: 7QB9) [37]. Sites in the N-terminal region where introduced cysteine substitutions could be labeled by MTSEA-biotin are shown as red sticks. Sites in the N terminus that could not be labeled by MTSEA-biotin are shown as green and blue spheres for the two different monomers. Sites in the N-terminal region that were not tested for MTSEA-biotin labeling are shown in grey. Portions of Ste2p other than the N terminus are depicted using cartoon representation, colored green and blue, respectively, for the two monomers.

**Figure 4 biomolecules-12-00761-f004:**
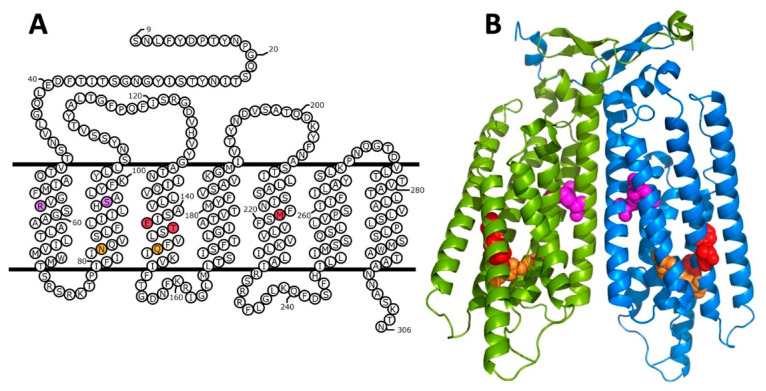
(**A**) Locations of mutation/suppressor pairs in relation to the transmembrane topology of Ste2p. Colored fills indicate different mutation/suppressor pairs. Red; loss of function mutations E143K and T144p are suppressed by M218T [54]. Orange; the Q149N mutation suppresses the constitutive activity of N84Q [56]. Magenta; various mutations at R58 suppress temperature sensitivity of S95Y [34]. (**B**) Locations of mutation/suppressor pairs in the cryo-EM structure of the ligand-free Ste2p dimer [37]. Colors of mutant/suppressor pairs are as shown in (panel **A**). Ste2p monomers are shown in blue and green.

**Figure 5 biomolecules-12-00761-f005:**
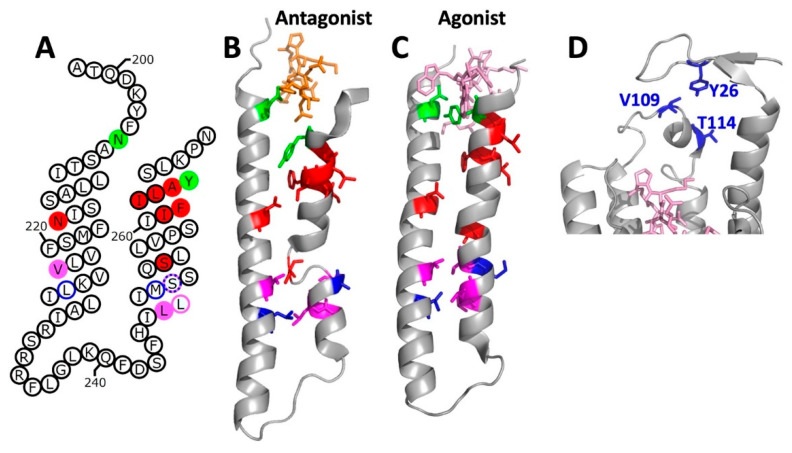
Locations of confirmed and putative disulfide crosslinks involving TM5 and TM6 (transmembrane segments 5 and 6). (**A**) Interactions identified by Lee et al. [57] are shown in green, interactions identified by Dube et al. [58] are shown in magenta and blue, and interactions shown by Taslimi et al. [59] are shown in red. In each case, disulfides confirmed by peptide mapping are indicated by colored fills. Putative interactions identified based on constitutive activation that was dependent on the presence of cysteine at paired sites are indicated by a colored circle around the indicated residue. (**B**) Locations of putative crosslinks mapped onto the three-dimensional structures of TM5 and TM6 in the antagonist-bound receptor structure (PDB Code: 7QA8). Colors are as shown in (panel **A**), but with no distinction between demonstrated functional and biochemically-confirmed interactions. Bound antagonist is shown in stick representation in orange. (**C**) Same as for (panel **B**), but mapped onto TM5 and TM6 of the agonist-bound receptor G protein complex (PDB code: 7AD3). Note that the cysteine replacement at residue S251 shows functional interactions with cysteines at the positions of either V223 or L226. This is indicated by alternating colored dashed circles in (panel **A**), but the residue is only colored as magenta in (panels **B** and **C**). Bound α-factor is shown in stick representation in pink. (**D**) Locations of putative crosslinks between cysteines substituted for Y26 in the N-terminal tail of Ste2p and for V109 and T114 in EC1. Relevant residues are shown in blue as stick representations.

**Figure 6 biomolecules-12-00761-f006:**
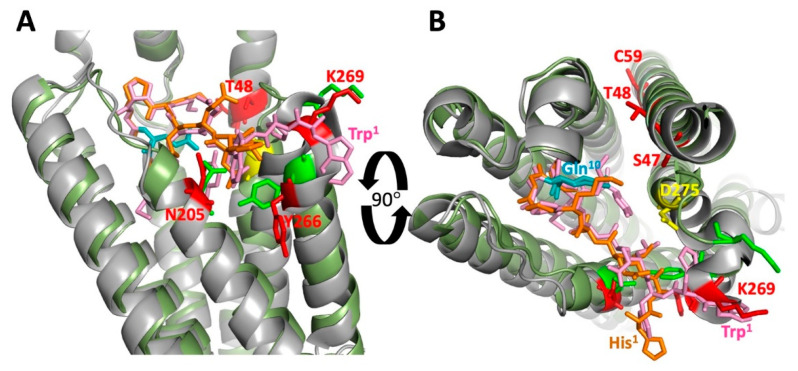
Locations in the three-dimensional structure of α-factor interactions with Ste2p identified in genetic and biochemical experiments. Panels (**A**,**B**) show views rotated 90 degrees with respect to each other. The antagonist-bound structure (PDB Code: 7QA8) is shown with the protein in cartoon representation in grey, and the antagonist in stick representation in orange. The agonist bound receptor-G protein complex is shown with the protein in olive green and α-factor in pink. Six residues in the protein that were previously reported to interact with the ligand are shown in stick representations in red (antagonist bound state) and green (agonist bound state; but only shown for residues N205, Y266, and K269, which change position between the agonist- and antagonist-bound structures). Residue Gln^10^ of the ligands is colored teal. Residue D275, proposed to undergo electrostatic interactions with bound agonist, is shown in yellow. The agonist- and antagonist-bound structures were superimposed using the “align” function of the PyMOL Molecular Graphics System (Schrödinger, LLC). In panel (**B**), the overlying regions of the receptor N-termini have been removed for clarity. The figure also clearly shows the altered position of residue Y266 in comparing the antagonist- vs. the agonist-bound receptor conformations, and the steric clash that would occur between Trp^1^ of α-factor and the extracellular end of TM6 in attempting to dock α-factor into the antagonist-bound conformation. It also shows that residue His^1^ of the bound antagonist occupies nearly the same position occupied by Trp^3^ of bound α-factor, a position ~9 Å removed from the position of the corresponding His^2^ residue of bound α-factor [37].

**Figure 7 biomolecules-12-00761-f007:**
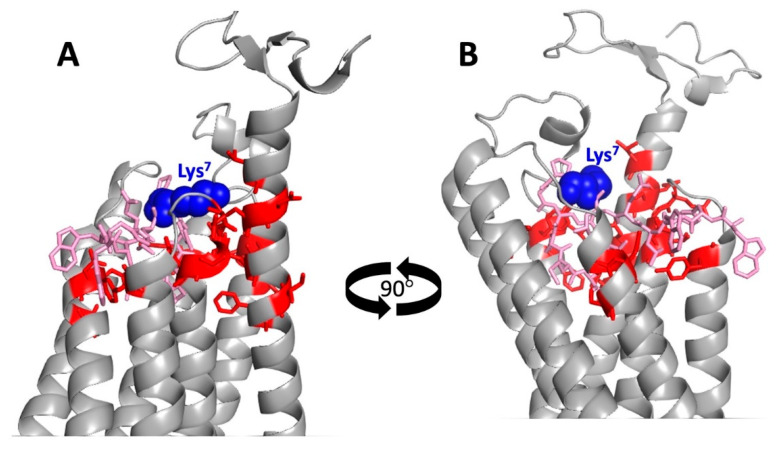
Locations in the Ste2p three-dimensional structure of residues that, when mutated, result in altered fluorescence emission of bound [Lys^7^(NBD),Nle^12^] α-factor [64]. Panels (**A**,**B**) show views rotated 180 degrees with respect to each other. Bound α-factor is shown as a pink stick representation with residue Lys^7^ shown as blue spheres. The sites of mutations that alter the fluorescence of the bound α-factor derivative are shown in red as stick representations. Note that the position of the NBD group conjugated to Lys-7 of α-factor is likely to be shifted compared to the indicated position of the native Lys^7^ side chain, since the bulkier derivative is sterically excluded from occupying the same location [77].

**Figure 8 biomolecules-12-00761-f008:**
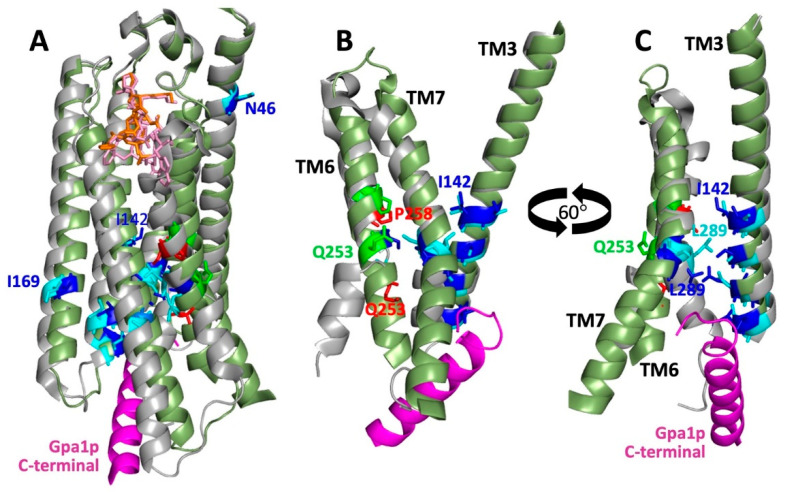
Locations of constitutively activating mutations in the structure of Ste2p (See Table 1). The aligned structures of antagonist- and agonist-bound Ste2p (PDB codes 7QA8 and 7AD3) are colored grey and olive green, respectively. Sites of the three strongest identified constitutive mutations, P258, Q253, and S259, [79,80,81] are shown in stick representations in red for the antagonist-bound structure, and in bright green for the agonist-bound structure. Other constitutive mutations listed in Table 1 are shown in stick representations in cyan for the antagonist-bound structure, and in dark blue for the agonist-bound structure. The C-terminal region of Gpa1p is shown in magenta. Panel (**A**) shows the entire Ste2p monomer with bound antagonist ([desTrp^1^Ala^3^Nle^12^]α-factor) shown in stick representation in orange, and bound agonist (α-factor) shown in pink. Panels (**B**) and (**C**) are two views, rotated 60° with respect to each other, showing just the sites of constitutive mutations on TM3, TM6, and TM7.

**Figure 9 biomolecules-12-00761-f009:**
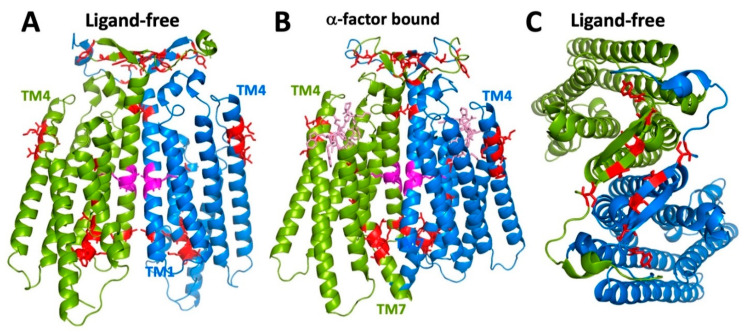
Locations of intermolecular disulfide crosslinks detected between Ste2p monomers. (**A**) Sites of crosslinks between introduced cysteine substitutions are shown mapped to the ligand-free Ste2p structure (PDB code: 7QB9) as red stick representations shown in a view parallel to the membrane. Ste2p monomers are shown in blue and green. The positions of residues constituting the GxxxG motif in TM1 implicated in contacts between monomers are shown as magenta sticks. (**B**) Sites of crosslinks between introduced cysteine substitutions are shown mapped to the structure of Ste2p in the activated receptor-G protein complex (PDB code: 7AD3) (coloring as in (panel **A**), with bound α-factor shown as pink sticks). (**C**) Sites of intermolecular crosslinks detected in the N-terminal tail of Ste2p shown in the structure of the ligand-free receptor (PDB code: 7QB9) viewed from the extracellular side of the membrane. Ste2p monomers are shown in green and blue. Sites of crosslinking in the N-terminal tail are shown as red sticks.

**Figure 10 biomolecules-12-00761-f010:**
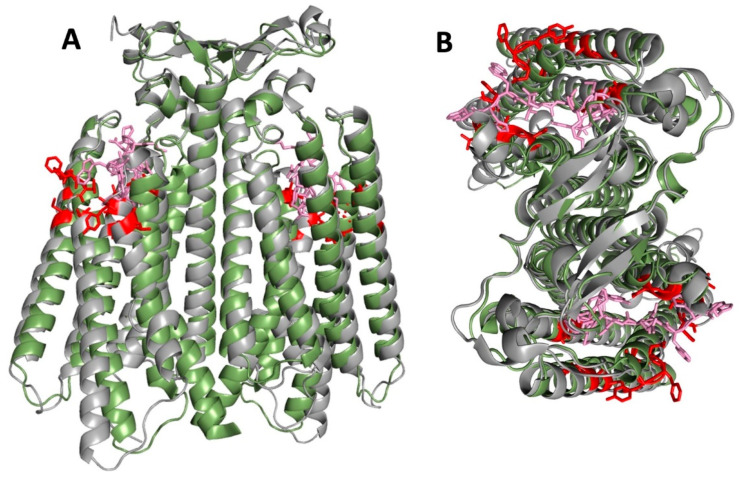
Locations of sites of dominant-negative mutations in Ste2p. (**A**) Sites of substitutions with dominant-negative effects are shown in red in stick representation mapped onto the ligand-free structure of Ste2p (PDB Code: 7QB9) shown in grey), viewed in a direction parallel to the membrane. The structure of the Ste2p in the α-factor-bound receptor-G protein complex is shown in olive green. Bound α-factor is shown as pink sticks. (**B**) Sites of dominant-negative mutations of Ste2p shown in a view from the extracellular side of the membrane. Coloring used is the same as that in panel (**A**).

**Table 1 biomolecules-12-00761-t001:** Mutations that result in constitutive activation of Ste2p.

Amino Acid Substitutions ^a^	Reference
N46S	[56]
**N84S**	[56,80]
S141P	[56]
I142T	[80]
S145L	[80]
**Q149R,V,P,T,A,S,I**	[56,80]
I150A	[56]
**I153F,Y,H,F,S,R,D,P,E**	[56]
I169K	[56]
L222P,R	[56,64]
L226W	[80]
**Q253L**	[78,80]
**S254F,L**	[78]
**P258M,Y,L,C, I,F,V,A**	[58,79,80,81]
**S259L,P**	[79,80]
S288P	[80]
L289S	[80]

^a^ This compilation does not include the weakest category of constitutive mutations identified by Parrish et al. [56]. The sites of the three strongest constitutive mutations identified by Sommers et al. [80], and the sites of the strongest category of constitutive mutations identified by Parrish et al. [56] are indicated in bold.

## Data Availability

Not applicable.

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
