# Peer review of "Comparison of Experimental Approaches Used to Determine the Structure and Function of the Class D G Protein-Coupled Yeast α-Factor Receptor"

_biomolecules, 2022, doi:10.3390/biom12060761_

Round 1
Reviewer 1 Report
This is a very nicely written review from leading experts on the topic of the pheromone receptor Ste2. The authors reinterpret prior genetic and biophysical analyses of Ste2 structure and function in light of recently published cryo-EM structures, including ligand-free, agonist-bound and antagonist-bound states. The conclusions from the cryo-EM structures are generally consistent with the results of previous approaches. To the extent that discrepancies exist, these can be interpreted either as illustrating limitations of the previous approaches or as evidence for the importance of dynamic functional properties of the receptor not captured by these static structures.
Major
I don’t feel like the second objective of the review has been adequately covered (“Can the comparison of the available structure with the results from the preceding genetic and biochemical approaches provide insights into dynamic properties and functional rearrangements of the structure that may not be evident from the cryo-EM analysis?”). I would like to hear from these experts about prospects for future studies that would be guided by the new cryo-EM structures. What aspects of the structural model (and it is still just a model) are yet to be tested, and if so how might these important questions be answered. What would future studies, those aimed at understanding dynamics, look like? Would they use variations on past approaches, or new approaches not yet implemented (NMR, EPR etc.)? Are there unique advantages to Ste2 for addressing these issues, and to what extent can anticipated results be extrapolated to other GPCRs? Adding a forward-looking paragraph on this would further strengthen an already excellent, timely, and clearly written contribution to the literature.
Minor
- 42 – “(ref)” needs citation
- 64 – Recent publications show that Gpa1 activates autophagy
- 65 – “(XXX)” needs citation
- 106 – the text suggests that the 7TM structure of GPCRs was already known when STE2 was first sequenced. The only sequence that came before Ste2 was rhodopsin. It was not appreciated until several years later that this is a topology shared by the GPCR family.
- 354 and l. 373 – is “normal agonist” the best term here? It suggests that there are abnormal agonists. Perhaps “physiological ligand/agonist”?
- 447 please explain how this observation explains the observe pH-dependent effects
- 528 please confirm that the authors mean “efficacy” and not “potency” (or both)
- 621 please confirm that the authors mean “inter-monomer contacts” and not “inter-molecular contacts” (or both)
- 702 – two models are proposed. Is there any reason to think that other mechanisms or other binding partners such as Sst2 could account to the dominant negative properties of these mutants?
there are a few grammatical errors and typos that need to be carefully checked, for example “One such hinge regions (sic) may involve contacts between TM6 and TM7 that involve several residues that provide constitutive activation when mutated. Another such region is the sting (sic) of contacts between the…”
Reviewer 2 Report
This review written by Dumont and Konopka reports that studies on the alpha-factor receptor Ste2p of the budding yeast Saccharomyces cerevisiae, which is one of the model organisms important for understanding diverse aspects of G protein-coupled receptor (GPCR) signaling. This paper is laudable and of interest to the fungal GPCR community. My evaluation is that the paper is publishable with only a few minor revisions.
1) Line 42. References are needed.
2) Line 65. I don’t understand the meaning of ‘XXX’.
3) Several sentences have no period. (e.g., Line 73, 280)
This manuscript should be thoroughly checked by authors before publication.
Author Response
Response to Reviewers
We thank the reviewer for their positive view of this manuscript and for their helpful comments and suggestions. Our responses are as follows:
Reviewer 2
Minor Comments:
1) Line 42. References are needed.
Changed as suggested
2) Line 65. I don’t understand the meaning of ‘XXX’.
Revised
3) Several sentences have no period. (e.g., Line 73, 280)
Changed as suggested
This manuscript should be thoroughly checked by authors before publication.
We have gone through the manuscript to correct additional typos and unclear sentences.